# Nuclear Organization in Stress and Aging

**DOI:** 10.3390/cells8070664

**Published:** 2019-07-01

**Authors:** Raquel Romero-Bueno, Patricia de la Cruz Ruiz, Marta Artal-Sanz, Peter Askjaer, Agnieszka Dobrzynska

**Affiliations:** Andalusian Center for Developmental Biology (CABD), Consejo Superior de Investigaciones Científicas/Junta de Andalucia/Universidad Pablo de Olavide, 41013 Seville, Spain

**Keywords:** chromatin, epigenetics, HGPS, laminopathies, longevity, mitochondrial unfolded protein response, progeria, NGPS, transcription

## Abstract

The eukaryotic nucleus controls most cellular processes. It is isolated from the cytoplasm by the nuclear envelope, which plays a prominent role in the structural organization of the cell, including nucleocytoplasmic communication, chromatin positioning, and gene expression. Alterations in nuclear composition and function are eminently pronounced upon stress and during premature and physiological aging. These alterations are often accompanied by epigenetic changes in histone modifications. We review, here, the role of nuclear envelope proteins and histone modifiers in the 3-dimensional organization of the genome and the implications for gene expression. In particular, we focus on the nuclear lamins and the chromatin-associated protein BAF, which are linked to Hutchinson–Gilford and Nestor–Guillermo progeria syndromes, respectively. We also discuss alterations in nuclear organization and the epigenetic landscapes during normal aging and various stress conditions, ranging from yeast to humans.

## 1. Introduction

All organisms are constantly exposed to a variety of stressors, including both environmental (such as temperature, pathogens, or nutrient availability) and internal factors (like cellular stress, mutations, or physiological aging). Furthermore, these stress-producing agents can interact, resulting in multiple direct and indirect effects on the organism. Several conserved mechanisms have evolved to counteract these effects, many of which rely on changes in nuclear organization and gene expression. In this review, we recapitulate recent findings on nuclear processes in the context of stress and aging.

The nuclear envelope (NE) acts as a physical barrier separating the nucleoplasm from the cytoplasm, but the NE is also a highly dynamic structure involved in maintenance of nuclear architecture, chromatin organization, DNA replication, regulation of gene expression, and signal transduction [1,2,3]. In reflecting upon the diversity and complexity of these processes, hundreds of proteins are reported to associate with the NE [1,4]. The NE is composed of inner and outer nuclear membranes (INM and ONM, respectively), separated by a 30–50 nm wide perinuclear space. The INM and ONM are fused where nuclear pore complexes form transport channels between the cytoplasm and the nucleus. In metazoans, the INM is underlaid by a protein meshwork, called the nuclear lamina, composed by lamin intermediate filaments and transmembrane proteins that provide physical stability to the nucleus and anchor chromatin at the NE [5,6]. The nuclear lamina also connects with the linker of nucleoskeleton and cytoskeleton (LINC) complex. The LINC complex is formed by the interaction of INM SUN domain proteins and ONM KASH domain proteins and has crucial roles in nuclear positioning and nuclear migration [7].

Within the nucleus, DNA together with histones and non-histone proteins are packed into chromatin that is classified as either compact heterochromatin or open euchromatin. Generally, euchromatin is transcriptionally active, gene-rich, and enriched for acetylation and methylation of lysine residues 4 and 36 of histone H3 (H3K4 and H3K36) [8]. By contrast, heterochromatin is transcriptionally repressed, gene-poor, and correlates with the methylation of H3K9 and H3K27. Heterochromatin is either facultative or constitutive [8]. Thus, constitutive heterochromatin contains many repetitive elements, frequent H3K9 methylation, and remains transcriptionally silent, while facultative chromatin is enriched for methylated H3K27 and genes that are only expressed at specific moments, for instance, during development. Heterochromatin typically accumulates at the nuclear periphery, whereas euchromatin is positioned in the nuclear interior. Since changes in this distribution are a characteristic of aging and senescence [9], we open our discussion with an overview of the role of the NE as chromatin organizer, followed by a recapitulation of the current knowledge about how mutations in NE genes are responsible for the development of the accelerated aging syndromes. Finally, we will address how cellular stress and the process of normal aging affect nuclear organization, specifically concentrating on epigenetic modifications.

## 2. The Nuclear Envelope as Chromatin Organizer

The nuclear lamina is a major constituent of the NE and is composed of INM proteins, and in metazoans, of nuclear lamins [10]. The genomes of mammals encode four lamins: A, B1, B2, and C, where A and C are both produced by the *LMNA* gene through alternative splicing. All mammalian cells express at least one B-type lamin, whereas lamin A/C is mainly found in differentiated cells. By contrast, most invertebrates express only B-type lamins. Lamins contact DNA directly or indirectly via lamin binding partners, providing the nuclear lamina with an important role in chromatin regulation [11]. Downregulation of lamins leads to severe defects in chromatin organization and impairment of the mechanical properties of the nucleus [12,13].

Chromatin regions positioned close to the nuclear lamina are composed mostly of heterochromatin and are often flanked by insulator protein CTCF-binding sites. Several mechanisms are involved in the establishment and maintenance of this distribution. For instance, H3K9 methylation and the chromodomain protein 4 (CEC-4) are required for heterochromatin anchoring to the nuclear periphery in *Caenorhabditis elegans* embryos [14,15]. In differentiated intestinal cells in *C. elegans*, MRG-1/MRG14, which binds the active mark H3K36me, is required to maintain the segregation of heterochromatin at the nuclear envelope [16]. In human cells, A-type lamins can act as transcriptional repressors when targeted to promoters [17,18] and depending on local chromatin modifications, lamin A/C downregulation increases the accessibility of these promoters which, in turn, enhances transcriptional permissiveness [18].

Gene repositioning from the periphery to the nuclear interior upon gene expression activation has been demonstrated in several systems, including *C. elegans* early larval development [19], *Drosophila melanogaster* S2 cells [20], and mouse lymphocyte development [21]. Nevertheless, nuclear positioning and gene expression are not always coupled processes: the elimination of either CEC-4 or the two H3K9 methyltransferases MET-2 and SET-25 leads to widespread release of heterochromatin from the nuclear periphery in *C. elegans* embryos, but is only accompanied by a few changes in gene expression [14,15,22]. Similarly, association or release of chromatin regions from the nuclear lamina during in vitro differentiation of mouse embryonic stem cells correlates with changes in the expression of only a subset of genes [23].

Interestingly, studies across different cell types have shown the existence of facultative, cell type-specific, and constitutive, cell type-invariant, lamina-associated domains (LADs) that are enriched for silent chromatin [24]. However, LADs do not only consist of heterochromatic regions. Recent studies describe lamin A/C and B1 binding to euchromatin regions (eLADs) in mouse fibroblasts and epithelial cells undergoing epithelial–mesenchymal transition [25,26]. LADs do not only have impacts on the local tethering of chromatin association to the nuclear periphery. LADs also affect the 3D structure of the genome through modifying interactions among topologically associating domains (TADs) [27,28,29]. Interestingly, an exhaustive study from van Steensel and colleagues has recently shown that many, but not all, repressed promoters in LADs are activated when moved to a more neutral chromatin environment [30].

Due to the biophysical and mechanical properties of lamins, their absence results in severe changes in nuclear morphology. This is also a hallmark of lamin mutations that lead to diseases called laminopathies [31,32]. The majority of laminopathies map to the *LMNA* gene with a strong prevalence of autosomal dominant missense mutations. Laminopathies affect a wide range of tissues: the striated muscle as Emery–Dreifuss muscular dystrophy (EDMD); metabolic diseases as the metabolic syndrome (MS); peripheral neuropathy like Charcot–Marie–Tooth (AD-CMT); and accelerated aging disorders, where the Hutchinson–Gilford progeria syndrome (HGPS) is the most studied [33]. Generally, laminopathies caused by lamin A mutations affect LADs and 3D chromatin organization [34]. Conservation of many of the disease-linked residues across evolution positions invertebrates—such as *D. melanogaster* and *C. elegans*—as powerful genetic models for studies of laminopathies [35,36,37]. For example, ectopic expression of a lamin variant corresponding to a human *LMNA* EDMD-causing mutation leads to dominant retention of muscle-specific promoters at the nuclear periphery, causing an altered expression and reduced muscle function [38]. Mutations in lamin-binding partners can also lead to disorders. For example, EDMD is causatively linked not only to mutations in *LMNA*, but also to the genes encoding the NE proteins emerin, SYNE1, and SYNE2, whereas disruption of MAN1 is responsible for the development of Buschke–Ollendorff syndrome [37]. A common feature of the disease-related NE proteins described here is that they are expressed in most or all tissues, and yet they are mostly linked to syndromes affecting specific cell types. Several models have been proposed to explain this paradox, including tissue-specific co-factors, different mechanical characteristics of the affected tissues and tissue-specific chromatin organization [31,39]. In all models, alterations in interactions between NE proteins and chromatin are proposed to be fundamental to the development of disease, but further investigations are required to determine the affected loci and precise molecular mechanisms.

The association of lamins and several other NE proteins with chromatin involves multiple interaction partners, such as barrier to autointegration factor 1 (BANF1 or BAF) and heterochromatin protein 1 (HP1). BAF is a highly conserved chromatin binding protein [40] required for mitotic chromosome coherence and nuclear assembly [41,42,43]. The localization of BAF at the NE is interdependent of its interaction partners, including lamins and emerin, but BAF is also present in the nucleoplasm [44,45]. Homodimers of BAF can bind simultaneously to DNA, lamins, and INM proteins harboring a LEM (LAP2, emerin, MAN1) domain, thus serving as a bridging factor [41]. Recent work found that BAF is involved in sensing mechanical stimuli to regulate cell cycle progression and DNA replication [46]. Interestingly, a single amino acid substitution of BAF in humans (Ala12Thr) causes Nestor–Guillermo progeria syndrome (NGPS; see below) [47].

Heterochromatin protein 1 (HP1) is part of a protein family largely conserved across eukaryotes [48,49]. HP1 was discovered in fruit flies as a gene silencing mediator localized to heterochromatin regions [50]. HP1 proteins consist of a chromodomain in the N-terminal part, essential for H3K9me2/3 binding, a C-terminal chromoshadow domain required for homodimerization, and a central hinge region which binds to nucleic acids and interacts with other chromosomal proteins to control chromatin architecture [48,49]. The conserved structure facilitated the identification of homologous proteins in many species, such as Swi6 in *Schizosaccharomyces pombe*; HPL-1 and HPL-2 in *C. elegans*; HP1a, b, and c isoforms in *D. melanogaster*; and HP1α, β, and γ isoforms in mammals. HP1 localizes throughout the nucleus but interacts specifically with lamin B receptor (LBR) and PRR14 at the NE [51,52]. Recently, HP1 has been linked to heterochromatin assembly though formation of phase-separated droplets, which compact the chromatin and promote the approaching of repressive elements. HP1 binding to DNA and the phosphorylation of its N-terminal extension are crucial for this phenomenon, which shows many common features with liquid phase separation [53,54,55].

Nuclear pore complexes (NPCs) are composed by multiple copies of ~35 different proteins termed nucleoporins (nups) and constitute transport channels between the nucleus and the cytoplasm [56]. Several nucleoporins are also engaged in contacts with the genome, either at NPCs or in the nuclear interior (reviewed in [57,58,59,60]). In yeast, the recruitment of chromatin to NPCs is generally associated with gene activation and transcriptional memory [60,61], whereas in organisms with larger genomes, many interactions are likely to involve nucleoporins travelling away from pores [59,62,63]. Recently, the relevance of deterioration of protein homeostasis during aging has been defined; specifically, the functional impairment of NPCs and other nuclear components [2]. Molecular turnover has a major impact on the nucleus, which experiences many specific alterations comparing long-lived cells with proliferating ones [2]. Pulse-chase experiments in the brain of rats found that 8–50% of the initial pool of nucleoporins and lamins were still present after 6 months (NUP93, NUP155, and NUP205 being particularly stable with 39–50% protein remaining; 13% in the case of lamin B1) [64]. Thus, it has been proposed that once NPCs are assembled into the NE, they are maintained for the rest of the life of the cell [64]. Moreover, in post-mitotic cells, the de novo expression of several nucleoporins is significantly reduced both in *C. elegans* and mice [65]. The lack of protein turnover may increase the level of oxidative damage of nucleoporins, thereby affecting NPC function and leading to aberrant distribution of nuclear and cytoplasmic macromolecules and loss of nuclear compartmentalization in aging cells [65].

## 3. Mutations in Nuclear Envelope Genes Causing Progeria

Progeria syndromes are devastating conditions that dramatically affect patients’ well-being and life expectancy (Table 1). The progression of symptoms suffered by progeria patients mimic, at least partially, the cellular and physiological alterations occurring during normal aging, implying that progeria models are relevant to studying general aging processes [66]. In the following paragraphs, we will describe progeria disorders in more detail.

Classical HGPS is caused by a missense de novo mutation in *LMNA* (c.1824 C>T) that creates a cryptic splice site leading to a deletion of 50 amino acids from the lamin A precursor protein. This deletion removes the cleavage site for the metalloproteinase ZMPSTE24 that is required to obtain mature lamin A protein. Instead, a permanently farnesylated and carboxymethylated lamin A form called progerin accumulates. Interestingly, progerin is also produced during normal aging, but in lower amounts than in HGPS [68]. Other LMNA mutations can cause either HGPS or atypical progeria syndromes, such as mandibuloacral dysplasia type a (MADA) [70]. Moreover, mutations in *ZMPSTE24* also lead to progeria syndromes, including restrictive dermopathy (RD) and mandibuloacral dysplasia type b (MADB) (Table 1) [73,74]. HPGS patients’ life expectancy barely reaches teenage years. The main cause of death is related to myocardial failure or stroke because of a rapid progression of atherosclerosis combined with other symptoms typical for aged people (alopecia, subcutaneous fat loss, bone and joint abnormalities, cachexia, etc.) that already appear a few months after birth.

Observations in patient fibroblasts indicate that the severity of HGPS might depend on the ratio between progerin and prelamin A accumulation [75]. Progerin expression and accumulation leads to a broad variety of phenotypes (Figure 1) characterized by aberrant nuclear morphology and architecture, genomic instability and transcriptional deregulation, telomere attrition, chromosome segregation defects, epigenetic alterations, stem cell exhaustion, impaired mechanotransduction, and altered intercellular communication (for a more detailed review of HGPS, see [32]), similar to what occurs in elderly people [76,77].

The expression of progerin modifies histone methylation patterns, specifically loss of H3K27me3, H3K9me3, and associated proteins like HP1, causing a reduction of peripheral heterochromatin [68,78,79] and altered CpG island methylation [80]. This affects the interaction of chromatin with nuclear lamins [81] and with other NE proteins like lamina-associated polypeptide-α (LAP2α) [26,82,83] and BAF [84]. Moreover, these proteins change their distribution due to the disruption of the nuclear lamina in HGPS cells. Accumulation of farnesylated progerin in HGPS cells also prevents phosphorylation of lamin A Ser22, leading to an accumulation of fibrous lamin A structure and affecting the nuclear shape and nuclear lamina functions and impairing its disassociation [85,86,87].

Recent studies have suggested novel therapeutic HGPS interventions. Screening an siRNA library for targets that affect progerin-induced phenotypes in cultured human fibroblasts led to the nuclear factor erythroid 2-related factor 2 (NFE2L2 or NRF2) antioxidant pathway [88]. The activity and localization of the NRF2 transcription factor is impaired in HGPS fibroblasts and NRF2 impairment recapitulates the progeroid phenotype. Strikingly, constitutively activated NRF2 ameliorates progerin-induced aging defects [88]. Smooth muscle cells are particularly vulnerable in HGPS patients. A recent study by Kim et al. reported that progerin expression induces apoptosis in smooth muscle cells when these are subjected to mechanical stress, thus mimicking the situation in tissues. Interestingly, LINC inhibition alleviates progeroid symptoms, in agreement with the LINC complex having a key role in mechanotransduction [89].

Telomere shortening is considered a hallmark of cell senescence and aging. Intriguingly, telomere damage also affects alternative splicing of numerous mRNAs, including *LMNA*, and thereby stimulates the expression of progerin [90]. Moreover, exogenous expression of telomerase in human fibroblasts prevented both senescence and progerin production, coupling the two processes [90]. Premature aging is linked to altered stem cell behavior and regenerative dysfunctions. Mesenchymal stem cells with either lamin A or WRN helicase mutations mimicking HGPS and Werner syndrome, respectively, have impaired stem cell proliferation and enter premature senescence due to heterochromatin loss [91,92].

A recessive mutation (Ala12Thr) in BAF was recently identified in two patients suffering from a related disease, coined the Nestor–Guillermo progeria syndrome (NGPS) [47]. Although NGPS shares phenotypic similarities with HGPS, including skeletal defects and scoliosis, cardiovascular deficiencies and metabolic complications characteristic of HGPS are absent. NGPS patients’ life expectancy is higher, and the manifestation of the illness occurs later than in HGPS, around 24 months after birth [93]. Similar to other laminopathies, nuclear morphology is abnormal in NGPS fibroblast, and can be restored by expression of wild type BAF [47]. It was proposed that the mutation destabilizes the BAF protein, although this has been questioned by others [94]. Paquet and co-workers reported that ectopically expressed mutant protein localizes correctly to the NE, but the mutation affects the DNA binding surface of BAF, thus possible interfering with chromatin anchoring at the NE [94].

The identification of mutations in BAF as causative of progeria, together with the accumulation of prelamin A isoforms during physiological and pathological aging, highlight the importance of NE components in human health. Thus, nuclear lamins, BAF, and other NE proteins would create a dynamic platform where the disruption of one element affects the localization or the function of the rest of elements, eventually causing alterations in peripheral chromatin anchoring and nuclear organization [84]. This is compatible with the spectrum of syndromes sharing phenotypic characteristics, even when the molecular causes are different [95].

Although the incidence is very rare, progeria requires intensive research efforts both for the improvement of the life quality of patients and for acquirement of knowledge potentially relevant to physiological aging. Indeed, many groups work on designing treatments for progeria disorders. Blocking the novel splicing site generated by the c.1824 C>T mutation in *LMNA* using a morpholino-based therapy restores nuclear morphology and extends lifespan in a HGPS mouse model [96]. Preventing prelamin A farnesylation using farnesyltransferase inhibitors rescues nuclear shape defects and shows improvement in bone structures and vascular stiffness both in a mouse model and in HGPS fibroblasts, where aberrant movement of interphase chromosomes is restored [97]. A clinical trial with a farnesyltransferase inhibitor (lonafarnib) in combination with other drugs is currently ongoing (https://clinicaltrials.gov/ct2/show/NCT00916747). However, inhibition of farnesylation does not ameliorate the DNA damage phenotypes unless the treatment is combined with other drugs such as rapamycin [98]. Another clinical trial combines farnesylation inhibition with a mTOR pathway inhibitor called everolimus, similar to rapamycin [99,100] (https://clinicaltrials.gov/ct2/show/NCT02579044). A potential concern regarding these protocols is that prelamin A and B accumulate in the nucleoplasm when farnesylation is inhibited, which may have a deleterious effect in terms of incorporation of mature lamin into the nuclear lamina [101]. Alternative prenylation of progerin might potentially also limit the efficiency of farnesyltransferase inhibitors [102]. For that reason, several studies have focused on other modifications of progerin, like reducing the activity of isoprenylcysteine carboxyl methyltransferase (ICMT) [103]. ICMT methylates farnesylated prelamin A during normal processing of prelamin A to lamin A. Interestingly, ICMT inhibition in *Zmpste24* mice suppresses NE accumulation of prelamin A and increases AKT–mTOR signaling, thereby enhancing cell proliferation and delaying cell senescence [103]. Several physiological parameters of *Zmpste24* mice are also improved when ICMT is inhibited, however, normal nuclear morphology is not restored [103]. Finally, recent studies have explored retinoids and vitamin D signaling as potential agents to reduce progerin expression in HGPS cells and thereby reverse their progeroid phenotype [104].

## 4. Alterations in Nuclear Organization during Aging

Aging is defined by degeneration in the cellular and molecular functions that culminate in cellular homeostasis wreckage. Long-term maintenance of nuclear architecture is vital for the normal functioning of cells and tissues over a lifetime [77]. During aging, many substantial nuclear changes are generated, such as alteration in nuclear shape (blebs, invaginations, and even fragmentation), redistribution of heterochromatin, and epigenetic modifications. These epigenetic changes comprise overall histone loss and alterations of covalent post-translational modifications, as well as nucleosome remodeling [77,105]. Despite the clear effect of chromatin alterations on the aging process, the detailed molecular mechanisms affecting longevity and aged-related phenotypes are still unclear [106]. While some chromatin modifications extend longevity in model systems (see Table 2), others lead to premature aging, as discussed below. Several recent studies have analyzed the correlation between the level and distribution of histone modifications and gene expression during aging [107,108,109]. For instance, in *Drosophila* male flies, increasing levels of H3K4me3 by mutation of little imaginal discs (*Lid*) histone demethylases result in shortened lifespan [110]. Furthermore, a similar effect occurs when inhibiting the homologous protein RBR-2 in *C. elegans*, leading to the interpretation that an excess of H3K4 trimethylation is unfavorable for lifespan [108]. However, defects in *Saccharomyces cerevisiae* Set1/COMPASS components (Swd1 or Spp1) responsible for the methylation of H3K4 broadly alter gene expression and reduce replicative lifespan [111]. Thus, either decreasing or augmenting H3K4 methylation can be detrimental to normal lifespan. Another histone modification, H4K20 trimethylation, is elevated in aged mammalian cells and correlates with impairment in the nuclear lamina [112,113], whereas loss of H3K27 methylation was observed at a subset of upregulated renal genes in aged rats [114].

Histone acetylation has a major role in the regulation of gene expression and chromatin organization, and has been intensively studied in the context of aging [126]. For instance, budding yeast undergoing replicative aging shows decreased levels of H3K56ac. In agreement with this, mutation of histone deacetylases *HST3* and *HST4*, which act on H3K56ac, reduces longevity [127,128]. A decline of the Sir2 NAD^+^-dependent histone deacetylase, which targets H4K16ac, as well as general histone depletion have been described in subtelomeric sites of replicatively old yeast cells [127]. Conversely, overexpression of Sir2 extends lifespan, thus acting as a limiting element in this process [120]. Recently, Li et al have confirmed, by microfluidic technologies, the role of Sir2 in the coordination of the different temporal patterns of heterochromatin de-repression during aging. This uncovered that sporadic silencing is important for lifespan regulation, whereas continued silencing or continued loss of silencing both decrease longevity [129]. Moreover, Larson and co-workers manipulated the levels of heterochromatin in *Drosophila* through regulation of HP1 and found that increased heterochromatin formation extended lifespan and reduced ribosomal RNA transcription [130]. The *C. elegans* Sir2 ortholog, SIR-2.1, is required for NAD^+^-dependent regulation of longevity via activation of the mitochondrial unfolded protein response (UPR^mt^; see below) [131,132], whereas its role in chromatin organization has not been explored. However, electron microscopy imaging found that peripheral heterochromatin is lost in old nematodes, concomitantly with changes in nuclear morphology [133,134], which also links chromatin organization with aging in this species. 

The nucleosome remodeling and deacetylase (NuRD) complex is a major chromatin remodeling complex implicated in regulation of nucleosome positioning and histone deacetylase activities [135,136]. This in turn affects transcription and the DNA damage response. During physiological and premature aging, NuRD complex function is compromised due to a decline of several of its subunits, which, together with the reduction of histone deacetylase 1 (HDAC1), increases the vulnerability of DNA to damage [137]. Finally, a neurological role of histone acetylation has been demonstrated in mice. In the brain of old mice, histone H4K12 is hypoacetylated, resulting in memory decline that can be reversed by restoring acetylation levels [138,139].

Another interesting observation related with chromatin structure and the process of aging is the progressive loss of histones. In yeast, during replicative aging there is a ~50% reduction in nucleosome occupancy along the entire genome that leads to transcriptional deregulation. This results in the expression of genes that are repressed in younger cells [140]. Loss of histones in an age-dependent manner has also been reported in *C. elegans* and human fibroblasts [141,142]. Finally, aging also impacts on chromatin composition through the exchange of histone variants. For instance, dynamic exchange of the conserved histone variant H3.3 is considered as a novel epigenetic mechanism [143]. Furthermore, accumulation of the H3.3 variant during aging results in changes to H3 methylation [144]. In *C. elegans*, expression of H3.3 increases during aging and this histone variant is required for stress resistance and enhanced lifespan in several long-lived mutants [145].

## 5. Stress and Epigenetics

In addition to aging and lifespan regulation discussed above, epigenetic mechanisms can modulate the cellular response to environmental stimuli, thereby ensuring genome stability and cell adaptation. The analysis of more than two hundred chromatin regulators and histone mutations in yeast brought to light a broad range of effects on gene expression patterns upon stress conditions, demonstrating an extensive role of epigenetics in stress response [66,146].

Environmental stressors have to be translated into intracellular signals, such as metabolic intermediates (e.g., acetyl CoA, *S*-adenosyl methionine, succinate, NAD^+^) and reactive oxygen species, in order to induce the proper cellular response [147]. Mitochondria provide an essential supply of several of these metabolic intermediates, which also serve as donors for epigenetic histone marks, such as acetylation and methylation. This highlights the direct relationship between metabolism and epigenome regulation [147] (Figure 2).

Lifespan extension upon mild mitochondrial stress is well described and conserved between species, for instance, through electron transport chain interference or defective ribosome function [153,154,155]. A common feature in mitochondrial enhancement of longevity seems to be the activation of the mitochondrial unfolded protein response (UPR^mt^), a conserved mitochondria-to-nucleus signal that triggers proteostasis and expression of metabolic genes to restore homeostasis [156]. Recently, an RNA interference-based screen in *C. elegans* identified the two histone lysine demethylases JMJD1.2/PHF8 and JMJD-3.1/JMD3 as regulators of lifespan in an UPR^mt^-dependent manner [150]. Expression of the demethylases correlates with a reduction of repressive H3K27me3 and results in UPR^mt^ induction both in nematodes and mice [150]. Also in *C. elegans*, Dillin and co-workers reported that mitochondrial stress induces global chromatin reorganization and decreases nuclear size [151]. Worms that were exposed transiently to mitochondrial stress during early development have an extended lifespan, suggesting that these changes are maintained into adulthood. Upon mitochondrial stress, the H3K9 methyltransferase MET-2 and its cofactor LIN-65 induce chromatin silencing but are proposed to also open specific chromatin regions, thereby allowing access for the UPR^mt^-related transcription factors DVE-1 and ATFS-1 [151].

As discussed above, sirtuins are a family of highly conserved histone deacetylases that regulate oxidative stress and lifespan. Under oxidative stress conditions, SirT1 is upregulated in mouse fibroblasts, which results in an increment of Suv39h1 methyltransferase half-life and, therefore, higher H3K9me3 levels [157]. Moreover, local loss of heterochromatin, which ultimately leads to the de-repression of genes that control the cell cycle, has been described in mouse neurons as a consequence of oxidative stress induced by overexpression of tau protein [158]. Acquisition of oxidative stress resistance is frequently influenced by chromatin proteins acting as epigenetic regulators. In *C. elegans*, stress resistance acquisition during development and its transmission across generations is regulated by the trithorax (WDR-5/ASH-2/SET-2) H3K4me3 regulatory complex that affects the output of multiple transcription factors, including DAF-16/FOXO, HSF-1, and SKN-1 [148,159,160]. Recent experiments suggest that this mechanism involves DNA methylation (N6-methyldeoxyadenine) of target genes by the DAMT-1 enzyme [152]. Moreover, a subunit of the trithorax complex in *S. cerevisiae*, Set4, acts as an oxidative stress protector through the activation of stress response genes [161]. Furthermore, the MYST family histone acetyltransferase complex contributes to oxidative stress resistance by H4K16 acetylation both in worms and humans, which results in the upregulation of FOXO transcription factors [121]. In agreement with these roles, inhibition of the trithorax complex in *Drosophila* reverses the increase in longevity and stress resistance conferred by mutations of the polycomb repressive complex 2 (PRC2) components E(Z) and ESC [123]. Opposing effects of polycomb and trithorax mutants are also seen when flies are exposed to desiccation stress but, remarkably, in this case, the former mutants are hypersensitive, whereas the latter show increased resistance [162]. Remarkably, heat stress-induced changes in the expression of an artificial multicopy transgene can persist for more than 10 generations in *C. elegans* through a mechanism that involves H3K9 methylation [163].

The global loss of heterochromatin and the alterations of H3K9me histone marks in many stress conditions have suggested a role for HP1 in the cellular stress response. For instance, HP1α and HP1γ are required to protect mouse embryonic fibroblast from genotoxic stress conditions through stabilization of the H3K9 methyltransferase Suv39h1 [164]. In *C. elegans*, heterochromatin mutants, among them *hpl-2*, show accumulation of DNA damage and genotoxic stress, which lead to sterility and growth impairment due to the activation of transposable elements [165,166]. However, *hpl-2* mutants also show altered lipid metabolism, increased longevity and higher endoplasmic reticulum stress resistance, suggesting that perturbation of heterochromatin organization has very pleiotropic effects and/or that HPL-2 might have additional functions other than heterochromatin organization [149,167].

Recently, a study of *D. melanogaster* HP1b mutants showed they have an increase in resistance to starvation and oxidative stress, decreased levels of mitochondrial activity, and alteration in fat levels [168]. Furthermore, under oxidative stress, the depletion of HP1 proteins generates a global loss of heterochromatin repressive marks, modifying the genome landscape to restore the homeostasis making a proper response to the stimuli [149,168]. Apparently, lack of HP1 proteins brings in stress resistance and global metabolism changes caused by de-repression of genes.

When organisms encounter conditions of insufficient or excess supply of nutrients, they change metabolic status, which frequently affects their lifespan. Several dietary stresses, especially the best characterized caloric restriction (CR), but also high-fat or low-protein diet result in epigenetic modifications that are conserved across evolution. In *S. cerevisiae*, nutritional stress leads to increased phosphorylation of histone H3 threonine 11 (H3pT11) by Sch9 and CK2 kinases, a change that is implicated in transcriptional activation and accelerates aging [169]. Accordingly, H3T11A mutants have extended chronological lifespan and increased stress resistance [169]. Also in yeast, a potential epistatic behavior was established between the loss of histone H4 N-terminal acetylation (caused by deletion of N-alpha-terminal acetyltransferase *NAT4*) and lifespan extension under CR conditions, since *nat4* deletion mutants have extended lifespan under normal growth conditions but not under CR [170]. In human lung fibroblasts, CR extends cellular lifespan by increasing the activity of SirT1, which results in repression of the p16/Rb pathway, blocking senescence and promoting cell cycle progression [171]. In *C. elegans*, the chromatin remodeling protein ZFP-1/AF10 is required to extend lifespan upon CR, at least partially through preventing the recruitment of FOXO transcription factors DAF-16 and PHA-4 to their target promoters [172]. Moreover, dietary restriction delays age-related DNA methylation changes in mice, reducing the expression of lipid metabolism genes in particular [173]. Interestingly, these changes are preserved across species (mice, rhesus monkeys, and humans) and observed throughout several tissues [174]. In mice, CR prevents the augmented levels of HDAC2 during aging in different hippocampus regions, which may delay aging-related phenotypes [175].

Diet-induced obese mice are characterized by multiple changes in histone modifications in the liver, white adipose tissue, and pancreas, including higher H3K36me2 and decreased H3K18ac and H3K23ac [176,177]. Interestingly changes in hepatic gene transcription induced by a high-fat diet were associated with increased levels of H3K27Ac, and were reversed after mice lost weight, suggesting highly dynamic control of the hepatic transcriptional landscape [178].

Finally, maternal dietary deficiency has been demonstrated to cause dramatic epigenetic modifications in the following generation. For example, female mice fed a low protein diet exhibit an increment of H3K9me3 levels at the promoters of *Cyp7a1* (cholesterol 7 alpha-hydroxylase) and *Igf2* (insulin-like growth factor-II), which cause a repression of these genes in their offspring. The latter is also accompanied by decrease of demethylase Jmjd2 during pregnancy [179,180]. Nuclear effects of stress also include global changes in chromatin folding. For instance, temperature stress in *Drosophila* cells causes general transcriptional silencing via polycomb, concomitantly with weakening of topologically associating domain (TAD) borders and increased frequency of inter-TAD interactions [181]. Very recently, similar effects were observed in human breast cancer cells upon mild hyperosmotic stress [182]. Changes in TAD organization were accompanied by alterations in binding profiles of cohesion and CTCF and were reversible: normal nuclear organization was re-established within 1 h after removal of the stress [182]. Finally, recent studies in *C. elegans* demonstrated a requirement of the H3.3 histone variant for proper response to high temperature stress in the soma and the germ line, since null mutants exhibit lower survival and smaller brood size compared to wild type, probably due to deficient induction of heat shock genes [145,183].

## 6. Conclusions

Despite many advances on the relationship between nuclear organization, stress, and aging, important questions remain, including how the interplay between cell organelles—such as the nucleus, mitochondria, and the endoplasmic reticulum—adjusts cellular responses upon exposure to different stressors. Growing evidence strongly associates changes in chromatin structure with organismal aging, and these modulations can impact on multiple nuclear processes. Changes in the distribution of histone modifications, decreased histone levels, and nucleosome remodeling that contributes to the opening of chromatin and transcriptional de-repression, are clearly related to the aging process and involve a wide variety of protein complexes, with histone methyltransferases, acetyltransferases, deacetylases, and chromodomain proteins representing just a few of the examples. Intriguingly, upon oxidative stress, abnormalities in histone modifications and loss of heterochromatin are observed, accenting the importance of the epigenetic modifications under stress conditions.

Alterations in the structure of the nuclear envelope by mutating nuclear lamins or lamina-associated proteins, like emerin or BAF, disrupt interactions between the nuclear lamina and peripheral chromatin. This leads to both de-repression of heterochromatin and changes in euchromatin, and can cause either tissue-restricted diseases or the development of systemic progeroid syndromes. Although it is still to be unveiled precisely how components of nuclear lamina associate with chromatin, the recent studies of many groups have concentrated on the more detailed characterization of gene regulation, specifically in the spatial organization of the genome. Thus, we expect that future research is likely to further expand our understanding of how the nuclear periphery and its individual components contribute to the formation of genomic compartments and influence gene expression. Understanding the global regulatory process controlling cellular shape, both under stress conditions and in the aging process, will facilitate the development of more efficient anti-aging strategies and therapies against progeria syndromes.

## Figures and Tables

**Figure 1 cells-08-00664-f001:**
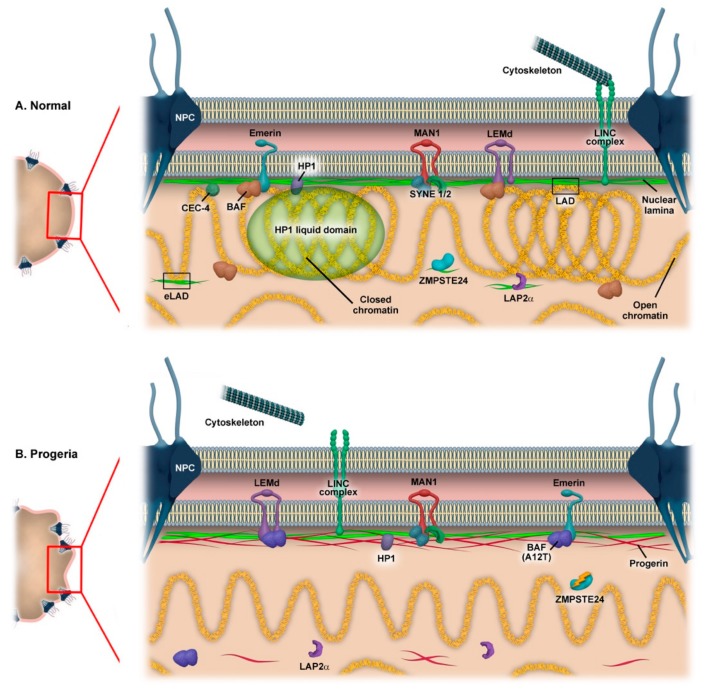
Schematic comparison of normal ((**A**) top) and progeria ((**B**) bottom) nuclei. In normal cells, repressed lamina-associated domains (LADs) accumulate at the nuclear periphery although lamin proteins also interact with euchromatin (eLAD). Other examples of nuclear envelope proteins involved in chromatin organization include BAF and chromodomain proteins, such as CEC-4 and HP1. The latter also promotes segregation of chromatin through phase separation. Linker of nucleoskeleton and cytoskeleton (LINC) complexes are responsible for nuclear positioning and transmission of mechanostimuli to the nucleus via the nuclear lamina. Other proteins are described in the main text. In progeroid cells, incorrect processing of lamin A (due to mutations in *LMNA* or *ZMPSTE24*) leads to accumulation of progerin, alterations in the structure of the nuclear lamina, and detachment of chromatin. Mutation in BAF can also lead to progeria, presumably through perturbation of BAF interaction with chromatin and/or lamins. Changes in nuclear morphology are frequently observed in cells from progeria patients. Moreover, reduced methylation of heterochromatin impairs contacts with HP1 and LAP2alpha.

**Figure 2 cells-08-00664-f002:**
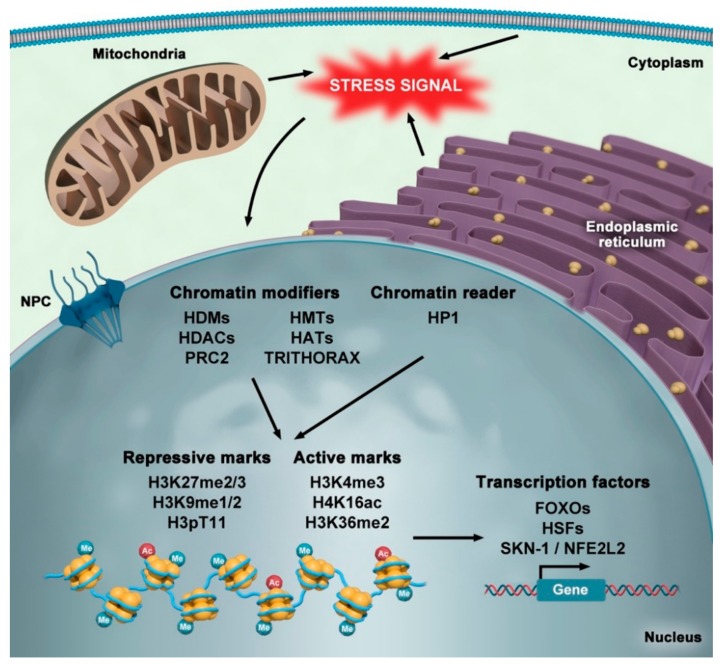
Epigenetic responses to stress. Stress signals from outside the cell or from specific organelles, such as mitochondria or the endoplasmic reticulum, are detected and processed by different signaling pathways [146]. The transduction of these stimuli results in the dynamic regulation of epigenetic mechanisms, mediated by a broad variety of chromatin protein modifiers acting as intermediates to make an appropriate cellular response [147]. Chromatin modifiers can add or remove specific histone modifications, such as methyl, acetyl, or phosphate groups. These different marks alter binding of chromatin readers (e.g., HP1) as well as gene expression through the action of different transcription factors (e.g., FOXOs, HSFs, SKN-1/NFE2L2) [121,123,148,149,150,151,152]. A non-exhaustive list of chromatin modifiers and readers associated to stress are depicted (abbreviations: HATs, histone acetyl transferases; HDACs, histone deacetylases; HDMs, histone demethylases; HMTs, histone methyltransferases; HP1, heterochromatin protein 1; NPC, nuclear pore complex; PRC2, polycomb repressive complex 2).

**Table 1 cells-08-00664-t001:** Progeria syndromes caused by mutations in nuclear envelope genes.

Disease		Gene	Reference
HGPS	Hutchinson–Gilford progeria syndrome	*LMNA*	[3]
APSs	Atypical progeria syndromes	*LMNA*	[67]
MADA	Mandibuloacral dysplasia	*LMNA*	[68]
RD	Restrictive dermopathy	*ZMPSTE24*	[69]
MADB	Mandibuloacral dysplasia	*ZMPSTE24*	[70]
NGPS	Nestor–Guillermo progeria syndrome	*BANF1*	[46]

Extensive overviews of additional diseases caused by mutations in genes encoding nuclear envelope components are found in [33,71,72].

**Table 2 cells-08-00664-t002:** Modifications in chromatin proteins associated with longevity. ^1^

Type of Enzyme	Protein	Intervention	Targets	Species	Reference
histone demethylase (HDM)	LSD-1	RNAi	H3K9me/ H3K4me	*C. elegans*	[115]
	UTX-1	deletion	H3K27me3	*C. elegans*	[116]
	SPR-5	deletion	H3K4me2	*C. elegans*	[117]
histone deacetylase/ remodeling complex (HDAC)	LET-418 (NuRD ^2^)	loss-of-function allele	Nucleosome	*C. elegans*	[118]
	SWI/SNF	RNAi	Nucleosome	*C. elegans*	[119]
	SIR2	overexpression	H4K16ac	*S. cerevisiae*	[120]
histone acetyltransferase (HAT)	MYS-1 & TRR-1	RNAi	H4K16ac	*C. elegans*	[121]
histone methyltransferase (HMT)	Trithorax (WDR-5, SET-2 and ASH-2)	deletion	H3K4me3	*C. elegans*	[108]
	SET-32	deletion	H3K9me3	*C. elegans*	[108,122]
	PRC2 ^3^	deletion	H3K27me3	*D. melanogaster*	[123]
	SET-18	deletion	H3K36me2	*C. elegans*	[124]
	SET-26	deletion	H3K4me3	*C. elegans*	[125]

^1^ All interventions in this table are associated with extended longevity, with the exception of inhibition of SWI/SNF, which shortens lifespan [119]; ^2^ Nucleosome remodeling and deacetylase; ^3^ Polycomb repressive complex.

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
