# Peer review of "Nuclear Organization in Stress and Aging"

_cells, 2019, doi:10.3390/cells8070664_

Round 1
Reviewer 1 Report
The review entitled: “Nuclear organization in stress and aging” by
Romero-Bueno et al., is clear, informative, well written and summarizing
a wide range of studies in a well-organized and logical manner.
Therefore I would recommend only few minor changes as described below
that I feel are needed for this review.
P2. Line 87 after the sentence citing ref 14, it will be beneficial to
add that in C.elegans there is an example in which affecting
heterochromatin anchoring to the nuclear periphery did not cause any
change in gene expression, cite ref 15 here too. Here and elsewhere in
the review I would be more cautious about the implications of
heterochromatin localization at the nuclear periphery on gene
expression, because the latter study (ref15) would rather suggest that
those are rather correlated features
P4 line 142. Gene silencing should be replaced by heterochromatin
assembly (which is not the same), and I would add the study of
Hiragami-hamada et al., 2016 in that paragraph.
P4 line 157. Please mention half-lives of NPCs and lamins in the text
with the related references.
P8. Table2 title maybe replace regulators with associated with
longevity. Add a legend explaining this table, the effect on the target
increase or decrease, add a column with the effect on longevity
(increase decrease)
P9 line 316 Typo in the sentence. P10. It is unclear whether figure 2
legend is more based on hypothesis or on evidence. Please provide the
refs in the figure legend or in an associated table.
P12. Line 415: I would replace arguing… by suggesting that hp-2 might
have might have additional function than heterochromatin organization.
Author Response
Reviewer #1
The review entitled: “Nuclear organization in stress and aging” by Romero-Bueno et al., is clear, informative, well written and summarizing a wide range of studies in a well-organized and logical manner. Therefore I would recommend only few minor changes as described below that I feel are needed for this review.
We thank the reviewer for her/his efforts and positive comments.
P2. Line 87 after the sentence citing ref 14, it will be beneficial to add that in C. elegans there is an example in which affecting heterochromatin anchoring to the nuclear periphery did not cause any change in gene expression, cite ref 15 here too. Here and elsewhere in the review I would be more cautious about the implications of heterochromatin localization at the nuclear periphery on gene expression, because the latter study (ref15) would rather suggest that those are rather correlated features
We have modified the text accordingly. We have also added a reference to Zeller et al, (Nature Genetics 2016), who showed de-repression of repetitive elements in set-25 met-2mutants.
P4 line 142. Gene silencing should be replaced by heterochromatin assembly (which is not the same), and I would add the study of Hiragami-hamada et al., 2016 in that paragraph.
We have modified the text accordingly and added the Hiragami-Hamada et al. reference, which indeed is very relevant to this argument.
P4 line 157. Please mention half-lives of NPCs and lamins in the text with the related references.
We added this information to the manuscript.
P8. Table2 title maybe replace regulators with associated with longevity. Add a legend explaining this table, the effect on the target increase or decrease, add a column with the effect on longevity (increase decrease)
We have modified the table accordingly.
P9 line 316 Typo in the sentence. P10.
We thank the reviewer for spotting this mistake. However, we decided to remove the sentence because it was not fitting well in the flow of the text.
It is unclear whether figure 2 legend is more based on hypothesis or on evidence. Please provide the refs in the figure legend or in an associated table.
We have added relevant references to the figure legend, which should clearify that the figure is based on evidence rather than hypotheses.
P12. Line 415: I would replace arguing… by suggesting that hp-2 might have might have additional function than heterochromatin organization.
We have modified the text accordingly.
Reviewer 2 Report
This is a good written review on a topic of general interest to readers of the journal Cells. No changes are required.
one minor point: line 409: A piont is missing at the end of the sentence.
Author Response
Reviewer #2
This is a good written review on a topic of general interest to readers of the journal Cells. No changes are required.
We thank the reviewer for her/his efforts and positive comments.
one minor point: line 409: A piont is missing at the end of the sentence.
We have modified the text accordingly.
Reviewer 3 Report
Comments
This is a comprehensive review that summarizes in a concise manner the data about the role of nuclear envelope proteins and histone modifiers in 3-dimentional organization of the genome. In addition, they also discuss alteration in nuclear organization and the epigenetic landscapes during normal aging and various stress conditions. This review article written by Raquel Romero-Bueno addresses the timely and exciting topic of the nuclear organization in stress and aging.
Minor points.
This manuscript would greatly benefit from the addition of a table summarizing diseases which are related to mutations of nuclear proteins and epigenetic modifiers (like table 2 for progeria).
Author Response
Reviewer #3
This is a comprehensive review that summarizes in a concise manner the data about the role of nuclear envelope proteins and histone modifiers in 3-dimentional organization of the genome. In addition, they also discuss alteration in nuclear organization and the epigenetic landscapes during normal aging and various stress conditions. This review article written by Raquel Romero-Bueno addresses the timely and exciting topic of the nuclear organization in stress and aging.
We thank the reviewer for her/his efforts and positive comments.
Minor points.
This manuscript would greatly benefit from the addition of a table summarizing diseases which are related to mutations of nuclear proteins and epigenetic modifiers (like table 2 for progeria).
We carefully considered this suggestion, but we realise that our review is already considerably longer that the average length of reviews in Cells so we prefer not to extend our manuscript further. We have added this text as legend to Table 1: “Extensive overviews of additional diseases caused by mutations in genes encoding nuclear envelope components are found in [33 + 71 + 72].”